# *mirror* determines the far posterior domain in butterfly wings

Martik Chatterjee[1,2]*[†], Xin Yi Yu[1][†], Noah K Brady[1], Connor Amendola[1], Gabriel C Hatto[1], Robert D Reed[1]*

[1]Department of Ecology and Evolutionary Biology, Cornell University, Ithaca, United States; [2]Depatment of Molecular Biology and Genetics, Cornell University, Ithaca, United States

*For correspondence:
mc2548@cornell.edu (MC);
robertreed@cornell.edu (RDR)

[†]These authors contributed equally to this work

Competing interest: The authors declare that no competing interests exist.

## eLife Assessment

This **important** study provides evidence of a deeply conserved role for the gene Mirror in providing positional identity in the posterior part of butterfly and fly wings, despite increased morphological complexity of butterfly wings. The findings are **solid** for the field of evo-devo. However, the tools in butterflies are more limited than in *Drosophila* and it is more difficult to determine which specific cells are mutant and whether the effect of mutation is cell-intrinsic. The work will be of interest to evolutionary and developmental biologists working on insect wing evolution and the evolution of patterning more generally.

**Abstract** Insect wings, a key innovation that contributed to the explosive diversification of insects, are recognized for their remarkable variation and many splendid adaptations. Classical morphological work subdivides insect wings into several distinct domains along the anteroposterior (AP) axis, each of which can evolve relatively independently to produce the myriad forms we see in nature. Important insights into AP subdivision of insect wings come from work in *Drosophila melanogaster*; however, they do not fully explain the diversity of AP domains observed across broad-winged insects. Here, we show that the transcription factor *mirror* acts as a selector gene to differentiate a far posterior domain in the butterfly wing, classically defined as the vannus, and has effects on wing shape, scale morphology, and color pattern. Our results support models of how selector genes may facilitate evolutionarily individuation of distinct AP domains in insect wings outside of *Drosophila* and suggest that the *D. melanogaster* wing blade has been reduced to represent only a portion of the archetypal insect wing.

## Introduction

Insect wings vary dramatically in their size and morphology. Butter`lies and moths in particular have attracted attention for their rapidly evolving wings that show extreme variation in shape and color pattern. Interestingly, the diversity of lepidopteran color patterns appears to follow relatively simple rules, wherein an evolutionarily conserved set of spot and stripe pattern elements can vary in size, color, and presence/absence along the anteroposterior (AP) axis (*Nijhout, 1991*; *Nijhout, 2001*). Surveys of wing pattern diversity across butterflies, considering both natural variation and genetic mutants, suggest that wings can be subdivided into five AP domains, bounded by the M1, M3, Cu2, and 2A veins as summarized in *Figure 1A* (*McKenna et al., 2020*). Within each domain, there is a strong correlation between pattern variation and wing morphology, and, conversely, between domains, there is a relative degree of independence in morphological variation. We know little about how these different AP domains are differentiated during wing development, however.

**Figure 1.** Proposed anteroposterior (AP) domains of insect wings. (**A**) Comparative morphology previously identified boundaries associated with color pattern variation, demarcated by the M1, M3, Cu2, and 2A veins (*de Celis et al., 1996*). (**B**) Examples of the 2A vein marking a clear posterior color pattern boundary across multiple butterfly families. (**C**) Proximal posterior features of the archetypal dipteran wing include the calypters, which are only found in the Calyptratae clade, and the alula, a lobe at the base of the wing blade (*McAlpine et al., 1981*). (**D**) *mirror* RNAi knockdowns result in highly specific loss of the alula in *Drosophila melanogaster* (*Kehl et al., 1998*). (**E**) Snodgrass' model of the archetypal insect wing specifies three major domains along the anteroposterior axis: the remigium, the vannus, and the jugum. Homologies of these domains with the features of the dipteran wing (**C**), or butterfly wing pattern boundaries (**A**), have remained unclear.

Some aspects of the AP wing differentiation process have been extrapolated from *D. melanogaster* wings to butterflies. The transcription factors engrailed and spalt define AP domains in developing fly wings (*Guillén et al., 1995*; *de Celis et al., 1996*; *Morata and Lawrence, 1975*) and exhibit expression patterns in butterflies that suggest some functions may be conserved. In butterflies, *engrailed* transcription occurs across the entire region posterior to the M1 vein, while its paralog *invected* marks a largely overlapping region between M1 and 2A (*Carroll et al., 1994*; *Keys et al., 1999*; *Banerjee and Monteiro, 2020*). *spalt* expression is more dynamic and complex in butterflies, but appears to consistently mark a region between the R2 and M3 veins (*Banerjee and Monteiro, 2020*). These observations, coupled with the overall size and morphological complexity of butterfly wings, have spurred interest in the question of whether additional AP differentiation mechanisms may underlie AP domain specification in Lepidoptera and other insects (*Abbasi and Marcus, 2017*; *Lawrence et al., 2019*). The identification of additional domain-determining transcription factors in broad-winged insects would provide a model for the modular diversification of insect wing morphology.

## Results and discussion

The goal of this study was to identify the molecular basis of AP domain specification in the butterfly wing, posterior to the M1 engrailed boundary. We initially identified the transcription factor *mirror* as a potential candidate selector gene based on previous mRNA-seq studies (*Hanly et al., 2019*), coupled with *D. melanogaster* work showing that *mirror* specifies the alula – a small membranous lobe at the posterior base of the wing in some dipterans (*Figure 1C and D*; *Kehl et al., 1998*; *Foronda et al., 2009*). We used in situ hybridization to visualize *mirror* mRNA localization in the last-instar imaginal discs of the common buckeye butterfly, *Junonia coenia*, and observed *mirror* expression throughout the wing domain posterior to the 2A vein boundary (*Figure 2A*; *Figure 2—figure supplement 1*). This expression precisely marks the vannus, or anal region (*Figure 1B and E*), of the butterfly wing – a distinct posterior domain of the wing blade populated by the anal (A) veins and bordered anteriorly

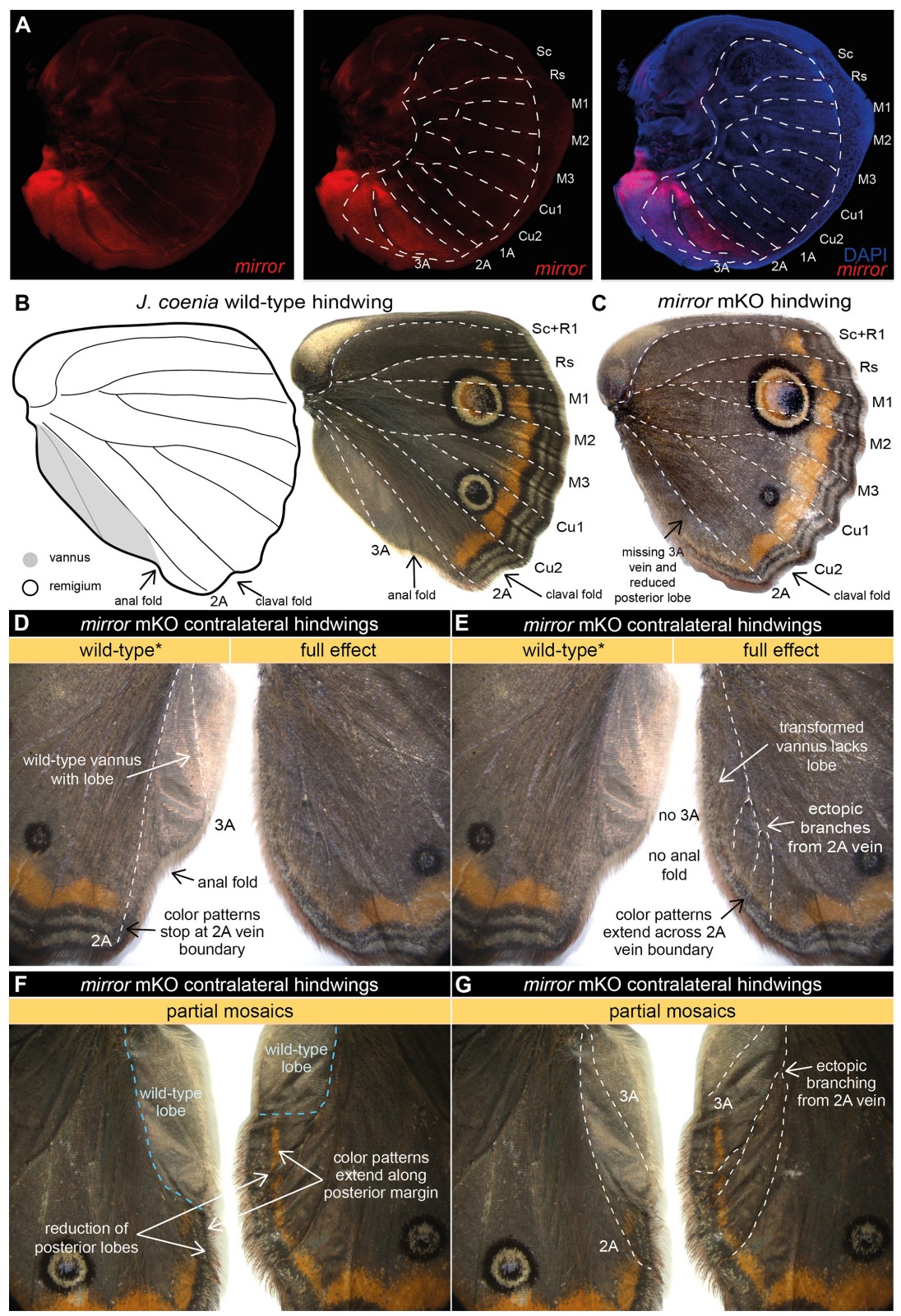

**Figure 2.** *mirror* determines the identity of the far posterior vannus domain in *J. coenia*. (**A**) In *J. coenia, mirror* (red) is expressed posterior to the 2A vein in the late last-instar hindwing imaginal discs, marking the region that will develop into the vannus. See *Figure 2—figure supplement 1* for forewing expression and controls. (**B**) In adult butterfly wings, the vannus is the wing field posterior to the 2A vein, shown here in *J. coenia* as a posterior lobe with silvery scales devoid of color patterns found anterior to 2A. (**C**) Targeted mosaic knockouts (mKOs) of *mirror* result in loss of vannus features

*Figure 2 continued on next page*

*Figure 2 continued*

in *J. coenia*. (**D**) Annotation of wild-type features preserved on one side of a contralateral mKO. (**E**) Annotation of full effect mutant phenotype in the opposing wing from the individual in (**D**) highlights loss of lobe, loss of 3A, ectopic branching of 2A, and posterior extension of color patterns. (**F**) Annotation of a partial mKO highlights transformation of posterior lobe and extension of color patterns, both of which terminate at the mutant clonal boundary. (**G**) The individual shown in (**F**) also shows ectopic branching of 2A in the mutant region, but preservation of 3A in the wild-type region. For additional *mirror* mKO phenotypes, see **Figure 2—figure supplements 2–4**.

The online version of this article includes the following figure supplement(s) for figure 2:

**Figure supplement 1.** Hybridization chain reaction (HCR) in situ hybridization of *mirror* highlights posterior domain expression in *J. coenia* wing imaginal discs.

**Figure supplement 2.** Mosaic knockouts (mKOs) of *mirror* result in partial or entire loss of posterior wing domain identity in dorsal *J. coenia* hindwings.

**Figure supplement 3.** Mosaic knockouts (mKOs) of *mirror* result in partial or entire transformation of the posterior wing domain.

**Figure supplement 4.** *mirror* mosaic knockouts (mKOs) show posterior margin anomalies in *J. coenia* forewings.

by the claval fold between the cubital (Cu) and A veins in the hindwing (**Figure 2B**). As observed in various representative species across different butterfly families, this region of the hindwing is distinct from the rest of the wing; the color patterns defining the wing blade terminate at the 2A vein, and in many species, it is marked by scales that differ in morphology and lack color patterns found in anterior regions of the wing (**Figures 1B and 2B**; **Figure 2—figure supplement 2A**). This region has been proposed to represent the posterior-most color pattern domain in butterflies (**Figures 1A and 2B**; **McKenna et al., 2020**).

To functionally assess the role of *mirror* in wing development, we generated CRISPR/Cas9 mosaic knockouts (mKOs) targeting the coding region of *J. coenia mirror*. We recovered many mKO mutants that consistently exhibited several distinctive phenotypes (**Figure 2C, D, and F**; **Figure 2—figure supplement 2**; **Figure 2—figure supplement 3**; **Figure 2—figure supplement 4**), but only ever in the *mirror*-expressing posterior vannus region:

1. Venation: In mKO mutant hindwings, we observed frequent truncation or complete loss of the 3A vein, as well as discontinuity and ectopic bifurcation of 2A veins in both hindwings (**Figure 2C–G**; **Figure 2—figure supplement 2**; **Figure 2—figure supplement 3**) and forewings (**Figure 2—figure supplement 4**).
2. Wing shape: mKOs consistently showed shape change along the posterior edge of the wing, especially the hindwing where the projecting lobe characteristic of the vannus was reduced and replaced by a more even, regular wing margin. The anal fold between the 2A and 3A veins, which marks the edge of the vannal lobe, was also lost in mutant clones affecting this wing region (**Figure 2C–G**; **Figure 2—figure supplement 2**; **Figure 2—figure supplement 3**).
3. Color pattern expansion: In mutant hindwings, we consistently observed the extension of wing margin color patterns across the 2A vein boundary into the posterior region of the wing. In wild-type wings, the submarginal and central symmetry system color patterns terminate at the 2A vein in both the dorsal (**Figure 2C–G**) and the ventral sides (**Figure 2—figure supplement 3**). Remarkably, in *mirror* mKO mutants, these entire color pattern systems extend past the 2A vein into the posterior regions of the wing on both dorsal and ventral wing surfaces. Specifically, submarginal bands continue past 2A to follow the posterior wing margin (**Figure 2C–G**; **Figure 2—figure supplement 2**), while central symmetry patterns extend from 2A down to the posterior margin (**Figure 2—figure supplement 3**).
4. Scale morphology: In most mutants vannus, scales lost their distinctive silver structural color and rounded apical edges and transformed into brown, serrated scales typical of the remigium (**Figure 2—figure supplement 2A**). We also observed that scales along mutant wing margins often displayed elongated scales, often manifesting as a dense hair-like mat across the posterior margin (**Figure 2—figure supplement 2A**).

The knockout phenotypes we observed illustrate several things about *mirror*'s function in wing development. First, across all mKO replicates, we never observed mutant phenotypes anterior to the 2A vein, which suggests that *mirror* effects are limited to the far posterior domain in which we observed *mirror* expression. Second, *mirror* knockouts appear to affect multiple aspects of wing morphology, including scale shape and coloration, overall wing shape, and wing venation. Finally, the loss of *mirror* function resulted in transformation, not loss, of the posterior domain. Even in mKO individuals that appeared to have a mutant phenotype across the entire posterior region, the *mirror*-expressing cells

posterior to the 2A vein were not simply lost – the wing field continues well past the 2A vein. Instead, the identities of the post-2A scale cells changed in color and structure. In this respect, the expansion of multiple color pattern systems into the posterior domain of mutants was particularly compelling because the loss of central symmetry system and submarginal patterns in the post-2A vannus region is commonly observed in Lepidoptera (*Figure 1B*). The ability to reconstitute these complex color pattern elements in the posterior domain by knocking out a single transcription factor leads us to speculate that their patterning information may occur in a latent form that is masked by *mirror* effects. Together, our expression and knockout data suggest that *mirror* acts as a selector gene necessary to define the identity of the far posterior wing domain.

Identifying *mirror* as a vannus-specifying factor is of interest for a few reasons. First, it provides a molecular explanation for how different AP domains of the butterfly wing may be individuated to independently evolve their own shape and color pattern variations. Previous authors have proposed the existence of such individuated domains and speculated that they may be specified by selector genes (*McKenna et al., 2020*; *Abbasi and Marcus, 2017*). Our data provide experimental support for this model and now motivate us to identify additional factors that may specify other domain boundaries between the M1 and A2 veins.

Next, the role of *mirror* in specifying the vannus provides an important link between molecular developmental biology and Snodgrass' classical anatomical designations of the insect wing fields – i.e., the remigium, the vannus, and the jugum (*Figure 1E*; *Snodgrass, 1935*). The remigium and the vannus are dominant features of most insect wings, while the jugum is a smaller lobe-like feature that is reduced or lost in many clades. The remigium is the major anterior wing blade that is directly articulated by thoracic motor musculature to power flight and is typically divided from the vannus by a distinct wing fold (e.g. the 'anal fold' in butterflies). In many insect orders, the vannus is a prominent fan-like structure, which can be a greatly enlarged gliding surface in some hemimetabolous clades such as Orthoptera (crickets and grasshoppers), Dictyoptera (mantids and roaches), Phasmatodea (stick insects), and Plecoptera (stoneflies). Importantly, *mirror* knockdown in the hemimetabolous milkweed bug *Oncopeltus fasciatus* causes developmental defects in the claval and anal furrows in the posterior wing (*Fisher et al., 2021*), which leads us to infer that *mirror*'s role in determining the vannus may be deeply conserved in insects. This prompts us to speculate that the developmental individuation of the posterior domain by *mirror* facilitated the remarkable diversification of the vannus across the insects.

Finally, the function of *mirror* in determining the alula in *D. melanogaster* (*Kehl et al., 1998*) suggests that the alula may represent an evolutionarily reduced vannus. The ramifications of this are significant for reconstructing the history of the insect wing, because they suggest that the *D. melanogaster* wing blade is a lone remigium – only one-third of the archetypal insect wing (*Figure 1E*). The dipteran jugum appears to have been lost entirely, except perhaps as a membranous haltere cover (calypter) in the Calyptratae clade, as speculated by *Snodgrass, 1935*. Thus, while flies have been an important model for characterizing genes and processes that build insect wings, a more comprehensive understanding of the development and evolution of insect wings requires work in species that have wings more representative of the complete ancestral blueprint.

# Materials and methods

**Key resources table**

| Reagent type (species) or resource | Designation | Source or reference | Identifiers | Additional information |
|---|---|---|---|---|
| Gene (*Junonia coenia*) | *mirror* | This paper | V2 genome: JC_02269-RA; V3 genome: JC_g12219 | V2 genome is available on https://www.lepbase.org/. Email authors if http://lepbase.org/ is inaccessible. |
| Strain (*Junonia coenia*) | Lab based for many generations – derived from wild collected individuals | | | |

*Continued on next page*

*Continued*

| Reagent type (species) or resource | Designation | Source or reference | Identifiers | Additional information |
|---|---|---|---|---|
| Sequence-based reagent | *mirror* HCR v3 probes | This paper; Molecular Instruments | Lot #PRQ901 | |
| Sequence-based reagent | *wingless* HCR v3 probes | Molecular Instruments | Lot #PRG129 | |
| Sequence-based reagent | *mirror* sgRNA-1 | This paper; IDT | CRISPR-cas9 guide RNA 5'-GAATGGACTTGAACGGGGCA | |
| Sequence-based reagent | *mirror* sgRNA-2 | This paper; IDT | CRISPR-cas9 guide RNA 5'-AGAAACAGGGTCGATGATGA | |
| Sequence-based reagent | Genotyping Primer Forward | This paper; IDT | Genotyping primer | 5'-CGCTTGTGCCCACCTTAAAC |
| Sequence-based reagent | Genotyping Primer Reverse | This paper; IDT | Genotyping primer | 5'-GTATGGCTCGGGGGATTCTG |
| Commercial assay or kit | HCR v3 Amplifiers, Buffers | Molecular Instruments | | |
| Commercial assay or kit | DNA extraction | Omega Bio-tek | E.Z.N.A. Tissue DNA Kit | |
| Commercial assay or kit | PCR purification | Omega Bio-tek | MicroElute Cycle-Pure Kit | |
| Software, algorithm | Inference of CRISPR Edits | Synthego | RRID:SCR_0024508 | |

## Identity of *Iroquois* family gene *mirror* ortholog in *J. coenia*

We performed a reciprocal BLAST of amino acid sequences of *D. melanogaster Iroquois Complex* genes *araucan*, *caupolican*, and *mirror* to identify their orthologs in the latest *J. coenia* and *Heliconius erato lativitta* genome sequences on lepbase.org and *Tribolium castaneum* and *Apis mellifera* on NCBI. The amino acid sequences of the top hits – from *J. coenia* (JC_02265-RA, JC_02269-RA), *H. erato lativitta* (HEL_009655-RA, HEL_009656-RA), *A. mellifera* (LOC412840, LOC412839), and *T. castaneum* (LOC652944, LOC660345) were aligned to the *Iroquois* genes of *D. melanogaster* using MUSCLE (**Edgar, 2004**). We used these alignments to build a maximum likelihood gene phylogeny on IQ-TREE 2 (**Minh et al., 2020**) using ModelFinder (**Kalyaanamoorthy et al., 2017**) for estimating the most accurate substitution model for our amino acid sequences. We used FigTree to visualize the tree depicted in **Supplementary file 1**.

## HCR fluorescent in situ hybridization of *mirror* mRNA

This protocol was adapted and modified from Bruce et al. (dx.doi.org/10.17504/protocols.io.bunznvf6). The coding sequence of JC_02269-RA was used by Molecular Instruments, Inc (Los Angeles, CA, USA) to design a *mirror*-specific hybridization chain reaction (HCR) probe set (Lot #PRQ901).

The probes are unique to the JC_02269-RA transcript as verified by BLAST search, using the latest *J. coenia* genome assembly (v2) on https://www.lepbase.org/, and do not share significant sequence similarity with any other transcripts, including the other *Iroquois* family member JC_02265-RA (see **Supplementary file 1**).

Last (fifth) instar larval wing imaginal discs were dissected in cold 1× PBS. The dissected discs collected from the left and right sides of the larvae were randomly assigned to control or treatment batches. Both control and treatment batches were fixed in 1× cold fix buffer (750 µL PBS, 50 mM EGTA, 250 µL 37% formaldehyde) on ice. The discs were then washed three times in 1× PBS with 0.1% Tween 20 (PTw) on ice, spending approximately 30 s to 2 min for each wash. Tissues were then gradually dehydrated by washing in 33%, 66%, and 100% MeOH (in PTw) for 2–5 min/wash on ice and stored in 100% MeOH at –20°C for days until the HCR protocol was started. On the day of the HCR protocol, the wing discs were progressively rehydrated in cold 75% MeOH, 50% MeOH, and 25% MeOH in PTw, spending 2–5 min/wash. After rehydrating, discs were washed once for

10 min and twice for 5 min with PTw. The discs were then permeabilized in 300–500 µL of detergent solution (1.0% SDS, 0.5% Tween, 50 mM Tris-HCl pH 7.5, 1 mM EDTA pH 8.0, 150 mM NaCl) for 30 min at room temperature. While waiting, hybridization buffer (Molecular Instruments) was warmed to 37°C (200 µL/tube). After permeabilizing with detergent solution, each tube of wing discs was incubated in 200 µL of pre-warmed hybridization buffer for 30 min at 37°C. Probe solution was prepared by adding 0.8 pmol (0.8 µL of probe from 1 µM stock solution) of each probe mixture to 200 µL of probe hybridization buffer at 37°C. The pre-hybridization buffer was removed, and the probe solution was added. For negative control tissues, hybridization buffer without probes was added instead. All wing discs were incubated overnight (12–16 hr) at 37°C. Before resuming the protocol, probe wash buffer (Molecular Instruments) was warmed to 37°C, amplification buffer (Molecular Instruments) was calibrated to room temperature, and a heat block was set to 95°C. The probe solution was removed and saved at –20°C to be reused. Wing discs were then washed four times in 1 mL pre-warmed probe wash buffer at 37°C for 15 min/wash. After the last wash step, discs were washed twice (5 min/wash) with 1 mL of 5× SSCT (5× sodium chloride sodium citrate, 0.1% Tween 20) at room temperature. The wing discs were pre-amplified with 1 mL of equilibrated amplification buffer for 30 min at room temperature. During this step, 2 µL of each of hairpin h1 and 2 µL of each hairpin h2 were mixed in 100 µL of amplification buffer at 95°C for 90 s and cooled to room temperature in the dark for 30 min. For *mirror*-only in situ hybridizations, we used B1 hairpin amplifiers tagged with Alexa Fluor 594 fluorophore (Molecular Instruments). For *mirror* and *wingless* double stains, we used *mirror* probes with B1 amplifier tagged with Alexa Fluor 647 fluorophore and *wingless* probes (Lot #PRG129) with B3 hairpin amplifiers tagged with Alexa Fluor 546 fluorophore. After 30 min, the pre-amplification buffer was removed from the discs, hairpin solution added, and the tissues were then incubated in the dark at room temperature for 2–16 hr. The hairpins were removed and saved at –20°C to be reused later. Excess hairpins were removed by washing five times (twice for 5 min, twice for 30 min, and once for 5 min) with 1 mL of 5× SSCT at room temperature. The wing discs were then incubated in 50% glycerol solution (in 1× PBS) with DAPI (0.01 µg/mL) overnight at 4°C. The wing discs were then mounted and visualized on a Zeiss 710 or Leica Stellaris 5 confocal microscope.

## CRISPR-Cas9-mediated mutagenesis of *mirror* in *J. coenia*

Two single-guide RNAs (sgRNAs; sgRNA1-5'-GAATGGACTTGAACGGGGCA; sgRNA2-5'-AGAAACAGGGTCGATGATGA) targeting the homeobox domain of *mirror* were mixed with 500 ng/µL of Cas9 nuclease and injected to *J. coenia* eggs 0.5–4 hr after oviposition (n=1042 eggs) as previously described (*Zhang and Reed, 2017*). Injected G0 individuals were reared on standard artificial diet until they emerged and were immediately frozen in –20°C upon emergence. Wings of surviving G0 adults (n=99) were assayed for anomalies to detect mKO phenotypes (n=29). Injection results and mutation phenotype frequencies are detailed in *Supplementary file 2*.

## Validating *mirror* mutants by genotyping

DNA was extracted using E.Z.N.A. Tissue DNA Kit (Omega Bio-tek) from the thorax of individuals that showed mutations in the wings. Extracted genomic DNA was amplified using a pair of primers flanking the sgRNA cut sites (Forward: 5'-CGCTTGTGCCCACCTTAAAC, Reverse: 5'-GTATGGCTCGGGGGGATTCTG). Amplified DNA was run on a 2% agarose gel and was excised and purified using the MicroElute Cycle-Pure Kit (Omega Bio-tek). Purified DNA was Sanger-sequenced by Cornell Institute of Biotechnology. Example mutant alleles of *mirror* are shown in *Supplementary file 3*.

## Acknowledgements

We thank the anonymous reviewers for their valuable suggestions and feedback. We thank Kate Siegel, Nigel Williams, and Rick Fandino for assistance with rearing butterflies; Johanna Dela Cruz for help with confocal microscopy at Biotechnology Resource Center (BRC) Imaging Facility (RRID:SCR_021741) at the Cornell Institute of Biotechnology. We also thank Dr. Eirene Markenscoff-Papadimitriou for using the Leica Stellaris 5 confocal microscope for imaging. This work was supported by the United States National Science Foundation grants NSF IOS-1753559 and IOS-2128164 awarded to RDR, and a Cornell Summer Experience Grant and an Office of Undergraduate Biology fellowship to XY.

## Additional information

### Funding

| Funder | Grant reference number | Author |
|---|---|---|
| National Science Foundation | IOS-1753559 | Robert D Reed |
| National Science Foundation | IOS-2128164 | Robert D Reed |

The funders had no role in study design, data collection and interpretation, or the decision to submit the work for publication.

### Author contributions

Martik Chatterjee, Conceptualization, Formal analysis, Supervision, Investigation, Methodology, Writing – original draft, Project administration, Writing – review and editing; Xin Yi Yu, Data curation, Formal analysis, Funding acquisition, Validation, Investigation; Noah K Brady, Connor Amendola, Gabriel C Hatto, Investigation, Visualization; Robert D Reed, Supervision, Funding acquisition, Project administration, Writing – review and editing

### Author ORCIDs

Martik Chatterjee (iD) https://orcid.org/0000-0003-4689-6212

Reviewer #1 (Public review): https://doi.org/10.7554/eLife.96904.3.sa1
Reviewer #2 (Public review): https://doi.org/10.7554/eLife.96904.3.sa2
Reviewer #3 (Public review): https://doi.org/10.7554/eLife.96904.3.sa3
Author response https://doi.org/10.7554/eLife.96904.3.sa4

## Additional files

### Supplementary files

Supplementary file 1. Identification of *mirror* ortholog in *J. coenia*. A maximum likelihood phylogeny of *J. coenia*, *H. erato lativitta*, *T. castaneum*, *A. mellifera*, and *D. melanogaster Iroquois Complex* genes confirms that JC_02269-RA is the ortholog of *mirror*.

Supplementary file 2. *mirror* CRISPR-Cas9 injection results.

Supplementary file 3. Sanger sequencing confirms mutations at CRISPR single-guide RNA (sgRNA) site in *mirror* 393 mosaic knockouts (mKOs). Sanger sequencing results from three different mirror mutant individuals. Each line denotes a 394 different wild-type (WT) or mutant allele with indels indicated in red and the PAM site in green.

MDAR checklist

### Data availability

The raw images, sequences used to generate gene tree and genotype sequence reads are submitted to Dryad: DOI: https://doi.org/10.5061/dryad.7sqv9s4xk.

The following dataset was generated:

| Author(s) | Year | Dataset title | Dataset URL | Database and Identifier |
|---|---|---|---|---|
| Chatterjee M, Yu XY, Brady NK, Hatto GC, Ammendola CA, Reed RD | 2025 | mirror determines the far posterior domain in butterfly wings | https://doi.org/10.5061/dryad.7sqv9s4xk | Dryad Digital Repository, 10.5061/dryad.7sqv9s4xk |

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
