## [Editor Report · eLife Assessment]

This **important** study provides evidence of a deeply conserved role for the gene Mirror in providing positional identity in the posterior part of butterfly and fly wings, despite increased morphological complexity of butterfly wings. The findings are **solid** for the field of evo-devo. However, the tools in butterflies are more limited than in *Drosophila* and it is more difficult to determine which specific cells are mutant and whether the effect of mutation is cell-intrinsic. The work will be of interest to evolutionary and developmental biologists working on insect wing evolution and the evolution of patterning more generally.

---

## [Referee Report · Reviewer #1 (Public review)]

Summary:

This short report shows that the transcription factor gene mirror is specifically expressed in the posterior region of the butterfly wing imaginal disk, and uses CRISPR mosaic knock-outs to show it is necessary to specify the morphological features (scales, veins, and surface) of this area.

Strengths:

The data and figures support the conclusions. The article is swiftly written and makes an interesting evolutionary comparison to the function of this gene in *Drosophila*. Based on the data presented, it can now be established that mirror likely has a similar selector function for posterior-wing identity in a plethora of insects.

Comments on revisions:

The revision is satisfactory. I agree with the authors that this article provides interesting insights on the evolution of insect wings. Of note, butterfly and fly wing imaginal disks differ in their mode of development: while fly wing disks grow as epithelial sacs that evaginate during metamorphosis, butterfly wing disks develop as relatively flat epithelial sheets that expand and differentiate progressively. This makes the similar role of mirror all the more interesting.

The revised text appropriately discuss how selector genes like mirror regionalize the wing during larval and pupal development. This article makes a reasonable use of CRISPR mosaic knock outs and uses contralateral controls to show the nature of the phenotypic transformations.

---

## [Referee Report · Reviewer #2 (Public review)]

This is a short and unpretentious paper. It is an interesting area and therefore, although much of this area of research was pioneered in flies, extending basic findings to butterflies would be worthwhile. Indeed, there is an intriguing observation but it is technically flawed and these flaws are far too serious to allow us to recommend publication

The authors show that mirror is expressed at the back of the wing in butterflies (as in flies). They present some evidence that is required for the proper development of the back of the wing in butterflies (a region dubbed the vannus by the ancient guru Snodgrass). But there are problems with that evidence. First, concerning the method, using CRISP they treat embryos and the expectation is that the mirror gene will be damaged in groups of cell lineages, giving a mosaic animal in which some lines of cells are normal for mirror and others not. We do not know where the clones or patches of cells that are defective for mirror are because they are not marked. Also, we do not know what part of the wing is wildtype and what part is mutant for mirror. When the mirror mutant cells colonise the back of the wing and that butterfly survives (many butterflies fail to develop), the back of the wing is altered in some selected butterflies. This raises a second problem: we do not know whether the rear of the wing is missing or transformed. From the images the appearance of the back of the wing is clearly different from wild type, but is that due to transformation or not? And then I believe we need to know specifically what us difference between the rear of the wing and the main part. What we see is a silvery look at the back that is not present in the main part, is it the structure of the scales? We are not told. There are other problems. Mirror is only part of a group of genes in flies and in flies both iroquois and mirror are needed to make the back of the wing, the alula (Kehl et al). What is known about iro expression in butterflies?

In flies, mirror regulates a late and local expression of dpp that seems to be responsible of making the alula. What happens in butterflies? Would a study of expression of Dpp in wildtype and mirror compromised wings be useful?

Thus, I find the paper to be disappointing for a general journal as it does little more than claim what was discovered in *Drosophila* is at least partly true in butterflies. Also it fails to explain what the authors mean by "wing domains" and "domain specification". They are not alone, butterfly workers in general appear vague about these concepts, their vagueness allowing too much loose thinking. Since these matters are at the heart of the purpose and meaning of the work reported here, we readers need a paper containing more critical thought and information. I would like to have a better and more logical introduction and discussion.

They do define what they mean by the vannus of the wing. In flies the definition of compartments is clear and abundantly demonstrated, with gene expression and requirement being limited precisely to sets of cells that display lineage boundaries. It is true that domains of gene expression in flies, for example, of the iroquois complex, which includes mirror, can only be related to pattern with difficulty. Some recap of what is known plus the opinion of the authors on how they interpret papers on possible lineage domains in butterflies might also be useful as the reader, is no wiser about what the authors might mean at the end of it!

The references are sometimes inappropriate. The discovery of the AP compartments should not be referred to Guillen et al 1995, but to Morata and Lawrence 1975.

Comments on revisions:

Nearly all the previous criticisms remain valid and are not discussed or overcome in the revision. The authors wish to draw their conclusions and we think they can do that, but they should make clear that key evidence is lacking. Thus their conclusions are speculative. But they present them more or less as facts. This is not justified. Let us suppose that clones lacking mirror do not survive or do not develop properly in the rear part of the wing and what they are seeing is occasional damage due to incomplete regeneration or to regenerative duplication?

Many clones in flies only include parts of one surface of the wing, could this happen here and how would it affect interpretations?

The null phenotype in the wing is not known but deduced from abnormal wings which "even in mKO..... appeared to have a mutant phenotype across the entire posterior region", a nice example of circular logic.

We believe the authors should be more objective and explain that their interpretations are not solid and that they should ideally be tested by finding ways of independently marking the clones. Other clonal mosaic experiments in butterflies have been done (eg https://journals.biologists.com/dev/article/150/18/dev201868/329659/Frizzled2-receives-WntA-signaling-during-butterfly) without cell autonomous independent markers, but they are more solid as transformed spots are made visible cell by cell by scale colour changes etc.

Their deduction that "mirror acts as a selector gene necessary to define the far posterior wing domain" is a speculative hypothesis, not a deduction and the readers should be so informed.

---

## [Referee Report · Reviewer #3 (Public review)]

Summary:

The manuscript by Chatterjee et al., examines the role of the mirror locus in patterning butterfly wings. The authors examine the pattern of mirror expression in the common buckeye butterfly, Junonia coenia and then employ CRISPR mutagenesis to generate mosaic butterflies carrying clones of mirror mutant cells. They find that mirror is expressed in a well-defined posterior sector of final-instar wing discs from both hindwings and forewings and that CRISPR-injected larvae display a loss of adult wing structures presumably derived from the mirror expressing region of hindwing primordium (the case for forewings is a bit less clear since the mirror domain is narrower than in the hindwing, but there also do seem to be some anomalies in posterior regions of forewings in adults derived from CRISPR injected larvae). The authors conclude that wings of these butterflies have at least three different fundamental wing compartments, the mirror domain, a posterior domain defined by engrailed expression, and an anterior domain expressing neither mirror or engrailed. They speculate that this most posterior compartment has been reduced to a rudiment in *Drosophila* and thus has not been adequately recognized as a such a primary regional specialization.

Critique: This is a very straight-forward study and the experimental results presented support the key claims that mirror is expressed in a restricted posterior section of the wing primordium and that mosaic wings from CRISPR injected larvae display loss of adult wing structures presumably derived from cells expressing mirror (or at least nearby). The major issue I have with this paper is the strong interpretation of these findings that lead the authors to conclude that mirror is acting as a high level gene akin to engrailed in defining a separate extreme posterior wing compartment. To place this claim in context, it is important in my view to consider what is known about engrailed, for which there is ample evidence to support the claim that this gene does play a very ancestral and conserved function in a defining posterior compartments of all body segments (including the wing) across arthropods.

(A) engrailed is expressed in a broad posterior domain with a sharp anterior border in all segments of virtually all arthropods examined (broad use of a very good pan-species anti-En antibody makes this case very strong).

(B) In *Drosophila*, marked clones of wing cells (generated during larval stages) strictly obey a straight anterior-posterior border indicating that cells in these two domains do not normally intermix, thus, supporting the claim that a clear A/P lineage compartment exists.

In my opinion, mirror does not seem to be in the same category of regulator as engrailed for the following reasons:

(1) There is no evidence that I am aware of, either from the current experiments, or others that the mirror expression domain corresponds to a clonal lineage compartment. It is also unclear from the data shown in this study whether engrailed is co-expressed with mirror in posterior most cells of J. coenia wing discs? If so, it does not seem justified to infer that mirror acts as an independent determinant of the region of the wing where it is expressed.

(2) The mirror is not only expressed in a posterior region of the wing in flies but also in the ventral region of the eye. In *Drosophila*, mirror mutants not only lack the alula (derived approximately from cells where mirror is expressed), but also lacks tissue derived from the ventral region of the eye disc (although this ventral tissue loss phenotype may extend beyond the cells expressing mirror).

In summary, it seems most reasonable to me to think of mirror as a transcription factor that provides important development information for a diverse set of cells in which it can be expressed (posterior wing cells and ventral eye cells) but not that it acts as a high level regulator as engrailed.

Recommendation:

While the data provided in this succinct study are solid and interesting, it is not clear to me that these findings support the major claim that mirror defines an extreme posterior compartment akin to that specified by engrailed. Minimally, the authors should address the points outlined above in their discussion section and greatly tone down their conclusion regarding mirror being a conserved selector-like gene dedicated to establishing posterior-most fates of the wing. They also should cite and discuss the original study in *Drosophila* describing the mirror expression pattern in the embryo and eye and the corresponding eye phenotype of mirror mutants: McNeill et al., Genes & Dev. 1997. 11: 1073-1082; doi:10.1101/gad.11.8.1073.

---

## [Author Response]

The following is the authors’ response to the original reviews

**Public Reviews:**

**Reviewer #1 (Public Review):**
Summary:This short report shows that the transcription factor gene mirror is specifically expressed in the posterior region of the butterfly wing imaginal disk, and uses CRISPR mosaic knock-outs to show it is necessary to specify the morphological features (scales, veins, and surface) of this area.Strengths:The data and figures support the conclusions. The article is swiftly written and makes an interesting evolutionary comparison to the function of this gene in *Drosophila*. Based on the data presented, it can now be established that mirror likely has a similar selector function for posterior-wing identity in a plethora of insects.

We thank the reviewer for their feedback.

Weaknesses:This first version has minor terminological issues regarding the use of the terms "domains" and "compartment".

We acknowledge that the terminologies “domains” and “compartments” might lead to confusion. To avoid confusion we have removed the term “compartment” from the manuscript.

**Reviewer #2 (Public Review):**
This is a short and unpretentious paper. It is an interesting area and therefore, although much of this area of research was pioneered in flies, extending basic findings to butterflies would be worthwhile. Indeed, there is an intriguing observation but it is technically flawed and these flaws are serious.The authors show that mirror is expressed at the back of the wing in butterflies (as in flies). They present some evidence that is required for the proper development of the back of the wing in butterflies (a region dubbed the vannus by the ancient guru Snodgrass). But there are problems with that evidence. First, concerning the method, using CRISP they treat embryos and the expectation is that the mirror gene will be damaged in groups of cell lineages, giving a mosaic animal in which some lines of cells are normal for mirror and others are not. We do not know where the clones or patches of cells that are defective for mirror are because they are not marked. Also, we do not know what part of the wing is wild type and what part is mutant for mirror. When the mirror mutant cells colonise the back of the wing and that butterfly survives (many butterflies fail to develop), the back of the wing is altered in some selected butterflies. This raises a second problem: we do not know whether the rear of the wing is missing or transformed. From the images, the appearance of the back of the wing is clearly different from the wild type, but is that due to transformation or not? And then I believe we need to know specifically what the difference is between the rear of the wing and the main part. What we see is a silvery look at the back that is not present in the main part, is it the structure of the scales? We are not told.

Thank you for this feedback. We appreciate that many readers may not accustomed to looking at mosaic knockouts. As discussed in a previous review article (Zhang & Reed 2017), we rely on a combination of contralateral asymmetry and replicates to infer mutant phenotypes. For many genes (e.g. pigmentation enzymes) mutant clones are obvious, but for other types of genes (e.g. ligands) clone boundaries are sometimes not directly diagnosable. It is simply a limitation of our study system. Nonetheless, you see for yourself that “the back of the wing is altered in some butterflies” – the effects of deleting *mirror* are clear and repeatable.

In terms of interpreting mutant phenotypes, we agree that that paper would benefit from a better description of the specific effects. Therefore, we have included an improved, more systematic description of phenotypes, along with better-annotated figures showing changes in wing shape and venation, scale coloration, and color pattern transformation (e.g. posterior elongation of the orange marginal stripes).

There are other problems. Mirror is only part of a group of genes in flies and in flies both iroquois and mirror are needed to make the back of the wing, the alula (Kehl et al). What is known about iro expression in butterflies?

In *Drosophila mirror, araucan*, and *caupolican* comprise the so-called *Iroqouis Complex* of genes. As denoted in Figure S4 and in Kerner et al (doi: https://doi.org/10.1186/1471-2148-9-74) the divergence of *araucan* and *caupolican* into two separate paralogs is restricted to *Drosophila*. As in most insects, butterflies have only two *Iroquois Complex* genes: *araucan* and *mirror*. We tested the role of *araucan* in *Junonia coenia* as shown in our pre-print: https://doi.org/10.1101/2023.11.21.568172. Its expression appears to be restricted to early pupal wings where it is transcribed in all scale-forming cells. Mosaic *araucan* KOs resulted in a change in scale iridescent coloration associated with changes in the laminar thickness of scale cells.

In flies, mirror regulates a late and local expression of dpp that seems to be responsible for making the alula. What happens in butterflies? Would a study of the expression of Dpp in wildtype and mirror compromised wings be useful?

We thank the reviewer for the proposal and agree that a future study comparing Dpp in wild-type versus *mirror* KO butterflies would be useful to clarify the mechanism of Dpp signalling in wing development. It is not clear, however, that the results of a Dpp experiment would change the conclusions of our current study therefore we decided not to undertake these additional experiments for our revision.

Thus, I find the paper to be disappointing for a general journal as it does little more than claim what was discovered in *Drosophila* is at least partly true in butterflies.

We respect that the reviewer does not have a strong interest in the comparative aspects of this study. Fair enough. This report is primarily aimed at biologists interested in the evolutionary history of insect wings.

Also, it fails to explain what the authors mean by "wing domains" and "domain specification". They are not alone, butterfly workers, in general, appear vague about these concepts, their vagueness allowing too much loose thinking.

A domain is “a region distinctively marked by some physical feature”. This term is used extensively in the developmental biology literature (e.g. “expression domain”, “embryonic domain”, “tissue domain”, “domain specification”) and is found throughout popular textbooks (e.g. Alberts et al. “The Cell”, Gilbert “Developmental Biology”). We prefer the term “domain” because of its association in the *Drosophila* literature with transcription factors that define fields of cells. We specifically avoided using the term “compartment” because of its association with cell lineage, which we have not tested.

Since these matters are at the heart of the purpose and meaning of the work reported here, we readers need a paper containing more critical thought and information. I would like to have a better and more logical introduction and discussion.

We would like the very same thing, of course, and we hope the reviewer finds our revised manuscript to be more satisfying to read.

The authors do define what they mean by the vannus of the wing. In flies the definition of compartments is clear and abundantly demonstrated, with gene expression and requirement being limited precisely to sets of cells that display lineage boundaries. It is true that domains of gene expression in flies, for example of the iroquois complex, which includes mirror, can only be related to patterns with difficulty. Some recap of what is known plus the opinion of the authors on how they interpret papers on possible lineage domains in butterflies might also be useful as the reader, is no wiser about what the authors might mean at the end of it!

We thank the reviewer for this suggestion. However, our experiments have little to contribute to the topic of cell lineage compartmentalization. We have therefore opted to avoid speculating on this topic to prevent confusion and to keep the manuscript focused on our experimental results.

The references are sometimes inappropriate. The discovery of the AP compartments should not be referred to Guillen et al 1995, but to Morata and Lawrence 1975. Proofreading is required.

We thank the reviewer for suggesting this important reference. We have included it in our revision.

**Reviewer #3 (Public Review):**
Summary:The manuscript by Chatterjee et al. examines the role of the mirror locus in patterning butterfly wings. The authors examine the pattern of mirror expression in the common buckeye butterfly, Junonia coenia, and then employ CRISPR mutagenesis to generate mosaic butterflies carrying clones of mirror mutant cells. They find that mirror is expressed in a well-defined posterior sector of final-instar wing discs from both hindwings and forewings and that CRISPR-injected larvae display a loss of adult wing structures presumably derived from the mirror expressing region of hindwing primordium (the case for forewings is a bit less clear since the mirror domain is narrower than in the hindwing, but there also do seem to be some anomalies in posterior regions of forewings in adults derived from CRISPR injected larvae). The authors conclude that the wings of these butterflies have at least three different fundamental wing compartments, the mirror domain, a posterior domain defined by engrailed expression, and an anterior domain expressing neither mirror nor engrailed. They speculate that this most posterior compartment has been reduced to a rudiment in *Drosophila* and thus has not been adequately recognized as such a primary regional specialization.Critique:This is a very straightforward study and the experimental results presented support the key claims that mirror is expressed in a restricted posterior section of the wing primordium and that mosaic wings from CRISPR-injected larvae display loss of adult wing structures presumably derived from cells expressing mirror (or at least nearby). The major issue I have with this paper is the strong interpretation of these findings that lead the authors to conclude that mirror is acting as a high-level gene akin to engrailed in defining a separate extreme posterior wing compartment. To place this claim in context, it is important in my view to consider what is known about engrailed, for which there is ample evidence to support the claim that this gene does play a very ancestral and conserved function in defining posterior compartments of all body segments (including the wing) across arthropods.(1) Engrailed is expressed in a broad posterior domain with a sharp anterior border in all segments of virtually all arthropods examined (broad use of a very good panspecies anti-En antibody makes this case very strong).(2) In *Drosophila*, marked clones of wing cells (generated during larval stages) strictly obey a straight anterior-posterior border indicating that cells in these two domains do not normally intermix, thus, supporting the claim that a clear A/P lineage compartment exists.In my opinion, mirror does not seem to be in the same category of regulator as engrailed for the following reasons:(1) There is no evidence that I am aware of, either from the current experiments, or others that the mirror expression domain corresponds to a clonal lineage compartment. It is also unclear from the data shown in this study whether engrailed is co-expressed with mirror in the posterior-most cells of J. coenia wing discs. If so, it does not seem justified to infer that mirror acts as an independent determinant of the region of the wing where it is expressed.(2) Mirror is not only expressed in a posterior region of the wing in flies but also in the ventral region of the eye. In *Drosophila*, mirror mutants not only lack the alula (derived approximately from cells where mirror is expressed), but also lack tissue derived from the ventral region of the eye disc (although this ventral tissue loss phenotype may extend beyond the cells expressing mirror).In summary, it seems most reasonable to me to think of mirror as a transcription factor that provides important development information for a diverse set of cells in which it can be expressed (posterior wing cells and ventral eye cells) but not that it acts as a high-level regulator as engrailed.Recommendation:While the data provided in this succinct study are solid and interesting, it is not clear to me that these findings support the major claim that mirror defines an extreme posterior compartment akin to that specified by engrailed. Minimally, the authors should address the points outlined above in their discussion section and greatly tone down their conclusion regarding mirror being a conserved selector-like gene dedicated to establishing posterior-most fates of the wing. They also should cite and discuss the original study in *Drosophila* describing the mirror expression pattern in the embryo and eye and the corresponding eye phenotype of mirror mutants: McNeill et al., Genes & Dev. 1997. 11: 1073-1082; doi:10.1101/gad.11.8.1073.

We thank the reviewer for their summary, critique, and recommendations. We agree with everything the reviewer says. Honestly, however, we were surprised by these comments because we took great care in the paper to never refer to *mirror* as a compartmentalization gene or claim it has a function in cell lineage compartmentalization like engrailed. As pointed out, we lack clonal analyses to test for compartmentalization. This is why we used the term “domain” instead of “compartment” in the title and throughout the manuscript. Nevertheless, we have recrafted the discussion in the manuscript, including completely removing the term “compartment”, to better avoid implications that *mirror* plays a role in cell lineage compartmentalization.

We also thank the reviewer for recommending the paper about the role of *mirror* in eye development. For the sake of keeping the paper focused, however, we decided not to broach the topic of *mirror* functions outside the context of wing development.

**Recommendations for the authors:**

**Reviewer #1 (Recommendations For The Authors):**
I have minor comments for improvement.The abstract and introductions are terminologically problematic when they refer to the concept of compartment and compartment boundaries. Allegedly this confusion has previously propagated in several articles related to butterfly wing development, which keeps alienating this literature from being taken seriously by fly specialists, for example. So it is important to use the right terms. I will try to explain point by point here, but I would appreciate it if the authors could undertake a significant rewrite taking these comments into account. The authors use the terms compartment and compartment boundary. This has a very specific use in developmental genetics: mitotic clones never cross a boundary (or compartment). I think the authors can keep referring to the equivalent of the A-P boundary, which is situated somewhere between M1-M2 based on unpublished data from the Patel Lab, and is not very well defined (Engrailed expression moves a little bit during development in this area). Domain is a looser term and can be used more liberally to describe genetically defined regions.- "Classical morphological work subdivides insect wings into several distinct domains along the antero-posterior (AP) axis, each of which can evolve relatively independently." Yes. This concept of domain and individuation seems important. You could make a proposed link to selector genes here.- "There has been little molecular evidence, however, for AP subdivision beyond a single compartment boundary described from *Drosophila melanogaster*." Incorrect, and this conflates "domain" and "compartment".Flies have wing AP domains too, that pattern their veins (see the cited Banerjee et al).- "Our results confirm that insect wings can have more than one posterior developmental domain, and support models of how selector genes may facilitate evolutionarily individuation of distinct AP domains in insect wings". Yes, and I like the second part of the sentence. Still, I would recommend simply deleting "confirm that insect wings can have more than one posterior developmental domain, and" because this is neglecting previous work on AP genetic regionalization in both flies (vein literature) and butterflies (e.g. McKenna and Nijhout, Banerjee et al).- "Analyses of wing pattern diversity across butterflies, considering both natural variation and genetic mutants, suggest that wings can be subdivided into at least five AP domains, bounded by the M1, M3, Cu2, and 2A veins respectively, within each of which there are strong correlations in color pattern variation and wing morphology (Figure 1A)". Yes, and I would recommend emphasizing they correspond to welldefined gene expression domains as mentioned in Banerjee et al, or McKenna and Nijhout.- "The anterior-most of these domains, bordered by the M1 vein, appears to correspond to an AP compartment boundary originally described by cell lineage tracing in *Drosophila melanogaster*, and later supported in butterfly wings by expression of the Engrailed transcription factor. Interestingly, however, *D. melanogaster* work has yet to reveal clear evidence for additional AP domain boundaries in the wing." Confusingly, because the first sentence is about compartments while the second is about AP domains. I also think the claim that Dmel has no other known AP domains is dubious because Spalt is highly regionalized in flies.- "Previous authors have proposed the existence of such individuated domains, and speculated that they may be specified by selector genes.5,10 Our data provide experimental support for this model, and now motivate us to identify factors that specify other domain boundaries between the M1 and A2 veins." Yes, I completely agree with this way to emphasize the selector effect, and to link it to the concept of "individuated domain"

We cannot thank the reviewer enough for the time and thought they devoted to giving helpful suggestions to improve our manuscript. We have applied all of the above recommendations to the revision.

Fig. S1: the field needs to move away from Red/Green microscopy images, for accessibility reasons.The easiest fix here would be to change the red channels to magenta.Green/Magenta provides excellent contrast and accessibility in general in 2-channel images.

We thank the reviewer for this suggestion. We have improved the color accessibility of Fig. S1.